# Color Stability Determination of CAD/CAM Milled and 3D Printed Acrylic Resins for Denture Bases: A Narrative Review

Mariya Dimitrova [1,*,†], Massimo Corsalini [2,*,†], Rada Kazakova [1,3], Angelina Vlahova [1,3], Giuseppe Barile [2,*], Fabio Dell'Olio [2,*], Zlatina Tomova [1], Stoyan Kazakov [4,‡] and Saverio Capodiferro [2,*,‡]

1 Department of Prosthetic Dentistry, Faculty of Dental Medicine, Medical University–Plovdiv, 4000 Plovdiv, Bulgaria; rada.kazakova@mu-plovdiv.bg (R.K.); angelina.vlahova@mu-plovdiv.bg (A.V.); zlatina.tomova@mu-plovdiv.bg (Z.T.)
2 Department of Interdisciplinary Medicine, 'Aldo Moro', University of Bari, 70100 Bari, Italy
3 CAD/CAM Center of Dental Medicine, Research Institute, Medical University–Plovdiv, 4000 Plovdiv, Bulgaria
4 Oral Surgeon, Private Dental Practice–Sofia, 1000 Sofia, Bulgaria; kazakovstoyan@gmail.com
* Correspondence: maria.dimitrova@mu-plovdiv.bg (M.D.); massimo.corsalini@uniba.it (M.C.); g.barile93@hotmail.it (G.B.); f.dellolio.odo@outlook.it (F.D.); capodiferro.saverio@gmail.com (S.C.)
† These authors contributed equally to this work.
‡ These authors contributed equally to this work.

**Abstract:** The aim of this paper is to review the available literature on the different methods for color stability determination of CAD/CAM milled and 3D printed resins for denture bases. The methodology included applying a search strategy, defining inclusion and exclusion criteria and selecting studies to summarize the results. Searches of PubMed, Scopus, and Embase databases were performed independently by three reviewers to gather the literature published between 1998 and 2022. A total of 186 titles were obtained from the electronic database, and the application of exclusion criteria resulted in the identification of 66 articles pertaining to the different methods for color stability determination of CAD/CAM acrylic resins for denture bases. Color change in dental materials is clinically very important for the dental operator, as it determines the clinical serviceability of the material. Discoloration of the denture bases can be evaluated with various instruments and methods. Dental resins may undergo color changes over time due to intrinsic and/or extrinsic factors. The extrinsic factors are considered the more frequent causes of color changes. According to a number of studies, CAD/CAM fabricated acrylics have achieved better color stability than the conventional PMMA (polymethyl methacrylate) resins.

**Keywords:** CAD/CAM dentures; digital dentures; milled acrylic resins; 3D printing; color stability

## 1. Introduction

Computer-aided design/computer-aided manufacturing (CAD/CAM) has become an increasingly popular part of dental medicine. There are numerous studies on the processing of acrylic materials utilizing this technology nowadays [1]. The result is more efficient planning of prosthetic CAD/CAM restoration, as this method can save materials, time and effort, and can even stimulate mass production. The technology involves designing the product from a three-dimensional (3D) file, imported in .STL (stereolithography, standard triangle language, standard tessellation language) format, which segments CAD drawings into small thin sections, thus allowing them to be processed layer by layer [2]. In recent years, with advancements in CAD/CAM technology, manufacturers have produced CAD/CAM PMMA (polymethyl methacrylate)-based polymer blocks as an alternative for denture base resins. The subtractive method for manufacturing dentures involves milling the denture base from a CAD/CAM PMMA disk (PMMA-disk) and then bonding a ready-made artificial tooth into the socket [3]. CAD/CAM PMMA block manufacturers

claim that these materials will have better mechanical properties than conventional denture base resins. These materials, which are polymerized under high temperature-pressure conditions, reduce residual monomer release, improve optical properties and improve the stability of color in comparison with the conventional acrylics. This type of dental resins facilitates the production of denture bases by easy milling [4]. Despite these advantages, the following disadvantages exist for PMMA: hypersensitivity, color changes over time, abrasion, and porosity [5]. Another limitation is the monochromatic and unesthetic teeth, which some manufacturers have overcome by using a unique layering system, resulting in polychromatic teeth that simulate the dentin and enamel of natural teeth, providing premium esthetics [6]. Color stability is one of the most important clinical properties for dental materials, and color change may be an indicator of aging or damaging of materials. Furthermore, the aesthetic appearance of a prosthesis is certainly an important feature required by patients and must satisfy their expectations. Evaluation of staining can be measured visually and by instrumental technique. Visual inspection of color evaluation is a psychologically and physiologically related issue. Instrumental technique greatly eliminates the chances of error during visual interpretation of color measurement [7]. The instrumental technique also eliminates the chances of subjective evaluation. Colorimeters and spectrophotometers are commonly used methods to evaluate change in color of dental materials.

### 1.1. Optical Properties of CAD/CAM Milled and 3D Printed Denture Base Resins

CAD/CAM dental resins are characterized by good optical properties; they transmit ultraviolet and visible light with a wavelength of up to 250 μm. According to the three-dimensional characteristic of colors, published by Alfred Munsell in 1905, the color space resembles a sphere in which each color occupies a specific place [8].

#### 1.1.1. Color Dimensions and Eye Perception

The physiology of color perception involves the following processes: light enters the eye through the cornea and the lens, whereafter the image is focused on the retina. The amount of light entering the eye is controlled by the iris, which expands or contracts depending on the level of illumination. Light is converted into nerve impulses, which are then transmitted to the cerebral cortex. The rods and cones are light-sensitive cells, the latter being responsible for clear vision and color perception [9].

- *Metamerism*

Light consists of different wavelengths, and the same tooth observed under different conditions will exhibit a different color, a phenomenon known as metamerism.

- *Transparency and translucency*

Transparency is a physical characteristic of material media, a measure of their permeability to electromagnetic waves or other types of radiation. How permeable an object is to radiation is determined by its characteristics in terms of reflection, scattering and absorption of that radiation [10]. Translucency is the relative amount of light transmission or diffuse reflection from a substrate surface through a turbid medium. Translucency was used to describe the optical properties of dental resin composites, ceramics, prosthetic elastomers, fiber posts, orthodontic brackets, natural tooth dentine and enamel, and combinations of materials [11].

Each color has three dimensions, including hue, saturation and brightness. Later, translucency was added as a fourth dimension, which represents the degree of transmitting passage of light through an object. When color is determined using the Munsell system, brightness is determined first, followed by saturation, and finally, hue. This system is used in the visual determination of color [12].

- *Hue (color name, color tone)*. It is dependent on the wavelength, determines the location of the color in the color space and is indicated by its name, red, green, etc., respectively.

- *Saturation (color density)*. The greater the amount of dye, the greater its degree. From a practical point of view, it is assumed that the color is richer, when its saturation is more significant.
- *Brightness (bleaching, color vitality)*. In Munsell's system, it is measured from 0 (pure black) to 10 (pure white). This indicator measures the amount of white, gray or black in a certain color. By reducing the black color, one goes through the different shades of gray. The spherical color space at the bottom of the sphere in the Munsell system is the magnitude of the bleaching (brightness, vitality) [13].

### 1.1.2. CIELAB Color System

In the objective measurements and comparisons of the colors of different objects, the system of the International Commission of Illumination (CIE), CIELAB, is used in the objective measurements and comparisons of the colors of different objects and works with Cartesian coordinates (Figure 1).

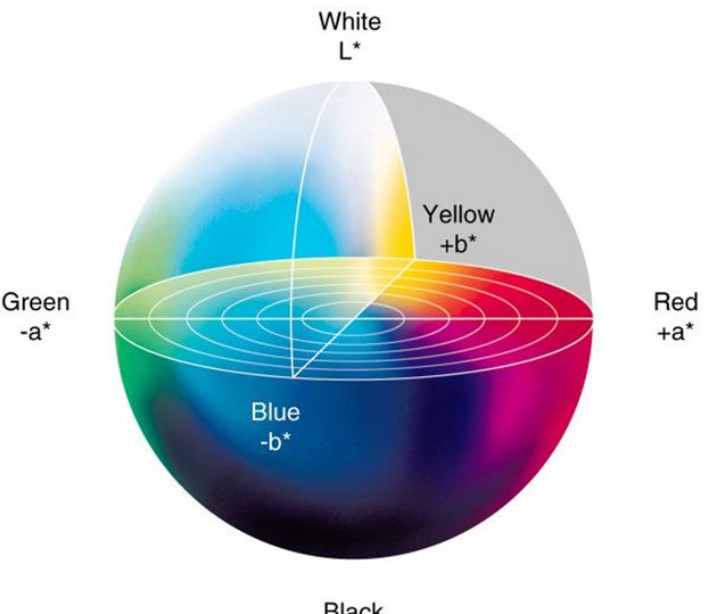

**Figure 1.** CIELAB color system.

The CIELAB color system expresses color as three values, L*, a* and b*, as the changes in these quantities represent a deviation along the respective axis. L* refers to lightness, and its value varies from 0 (for perfect black) to 100 (for perfect white). The a* axis is relative to the chromaticity of the red-green opponent colors. The b* axis is relative to the chromaticity of the yellow-blue opponent colors [14].

This system is used for objective and precise color determination in instrumental measurement. The location relative to the axes a* and b* determines the exact position of the color in a particular plane of the color space. The value L* characterizes the whitening of the color. With the help of this system, and in combination with the precision colorimeter, it is possible to make an objective numerical measurement of the color of various objects using a standard light source on a white background [15].

### 1.1.3. CIEHLC Color System

The "CIELCh" or "CIEHLC" space is a color space based on CIELAB, which uses the cylindrical coordinates C* (chroma, relative saturation) and h° (hue angle, angle of the hue in the CIELAB color wheel) instead of the Cartesian coordinates a* and b*. The CIELAB lightness L* remains unchanged [16]. (Figure 2).

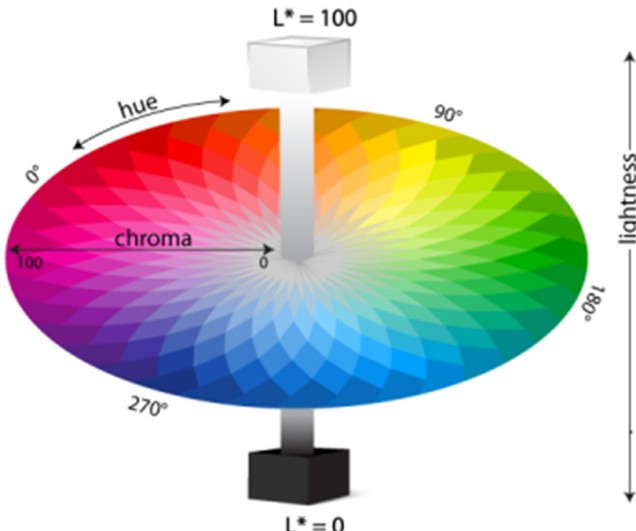

**Figure 2.** CIELCH Color System.

1.1.4. Conversion between CIELAB and CIEHLC

The conversion of a* and b* to C* and h° is performed as follows with (Equation (1)):

$$X = X_n f^{-1}\left(\frac{L^*+16}{116} + \frac{a^*}{500}\right)$$

$$Y = Y_n f^{-1}\left(\frac{L^*+16}{116}\right) \tag{1}$$

$$Z = Z_n f^{-1}\left(\frac{L^*+16}{116} + \frac{b^*}{200}\right)$$

Equation (1) is the formula of the conversion of a* and b* to C* and h° (CIELAB to CIEHLC).

*X, Y, Z* describe the color stimulus considered and Xn, Yn, Zn describe a specified white achromatic reference illuminant. The reverse transformation is most easily expressed using the inverse of the function *f* above.

$$L^* = 116\, f\left(\frac{Y}{Y_n}\right) - 16$$

$$a^* = 500\left(f\left(\frac{X}{X_n}\right) - f\left(\frac{Y}{Y_n}\right)\right) \tag{2}$$

$$b^* = 200\left(f\left(\frac{Y}{Y_n}\right) - f\left(\frac{Z}{Z_n}\right)\right)$$

Equation (2) is the formula of the conversion of C* and h° to a* and b* (CIEHLC to CIELAB).

*X, Y, Z* describe the color stimulus considered and Xn, Yn, Zn describe a specified white achromatic reference illuminant. For the CIE 1931 (2°) standard colorimetric observer and assuming normalization, where reference white = Y = 100, the values are as follows.

For standard illuminant D65,

$X_n$ = 950,489
$Y_n$ = 100
$Z_n$ = 1,088,840

For illuminant D50, which is used in the printing industry,

$X_n$ = 964,212
$Y_n$ = 100
$Z_n$ = 825,188

The LCh color space is not the same as the HSV, HSL or HSB color models, although their values can also be interpreted as a base color, saturation and lightness of a color [17]. The HSL values are a polar coordinate transformation of what is technically defined the RGB cube color space. LCh is still perceptually uniform. Furthermore, *H* and *h* are not identical, because HSL space uses as primary colors the three additive primary colors red, green, and blue (*H* = 0, 120, 240°). Instead, the LCh system uses the four colors red, yellow, green, and blue (*h* = 0, 90, 180, 270°) [18]. Regardless of the angle *h*, *C* = 0 means the achromatic colors, that is, the gray axis. The simplified spellings LCh, LCH and HLC are common, but the latter presents a different order. HCL color space (Hue-Chroma-Luminance), on the other hand, is a commonly used alternative name for the L*C*h (uv) color space, also known as the cylindrical representation or polar *CIELUV*. This name is commonly used by information visualization practitioners who want to present data without the bias implicit in using varying saturation [13,14]. The name Lch (ab) is sometimes used to differentiate from L*C*h (uv) [19]. CIELAB or CIE L*a*b* is a device-independent, 3D color space that enables accurate measurement and comparison of all perceivable colors using three color values. In this color space, numerical differences between the values roughly correspond to the amount of change humans observe between colors. This is one of the main reasons why the CIELAB color system is mostly preferred [20].

### 1.2. Objective Methods for Color Stability Evaluation of CAD/CAM Acrylic Resins

Esthetic considerations, including color accuracy, are an important aspect of dental prosthetics. In addition, the denture base material should match the color and appearance of the underlying tissues. One of the most important clinical features of all dental materials is color stability and any color changes are indicators of aging or damaged materials. The combination and duplication of tooth colors have been improved with advances in dentistry. There are several systems that include visual and instrumental methods for determining color [21]. These systems allow the examination of the appropriate porcelain or resins used for making removable dentures. The methods for determining the color can be divided into manual (visual) and instrumental (hardware). The first group of methods for determining optical changes involves visual assessment, which is quite subjective and raises the need for quantitative analysis of the data. This is the most commonly used method in dental practices [22]. In the second group of methods, we consider the device colorimeter, which is a measuring device that measures the absorption of certain light wavelengths by a specific dissolvent. The three-stimulus colorimeter is used for color calibration, as well as for comparative analysis. By the use of a spectrophotometer, a special device for measuring a given optical spectrum, the tests for registration of color changes are performed. Each of these methods has its advantages and disadvantages, and by utilizing a combined method, with instrumentation and visual assessment, the most reliable results are obtained [23].

### 1.2.1. Visual Methods for Color Determination

Dental professionals determine the optical changes in the oral tissues by comparing it to the chosen color, which contains the necessary set of optical standards. The choice of color is recommended to be made at the beginning of the patient's visit. It is possible to trace or supplement the color information at the next visit. There are several different shade guides that are optional for color determination of the denture bases. The Lucitone 199 Shade Guide is a valuable prescription tool for dentists when requesting top-quality denture base material. Four durable high-gloss uniquely shaped tabs are displayed on a convenient ring, featuring Lucitone199 esthetic shades, including original, light, light reddish Pink, and dark pink [24]. Oral tissues should be at the eye level; this is how the most color-sensitive part of the retina is used. The time for optimal color determination is about 15–20 s. If it takes longer to determine the color, it is recommended to shift the gaze

to another surface (preferably gray) [25]. Squinting also helps determine color saturation. Eye rest is performed by stopping the gaze for at least 30 s. This leads to a balance of color receptors in the retina, so that the eye can perceive again the predominantly yellow color shades of tooth tissue. If, however, there is a difficulty in determining the color, a color standard with a lower saturation value and greater brightness could be set. Thereafter, shape, function and comfort follow [26].

1.2.2. Instrumental Methods for Optical Changes Determination

- *Colorimeters*

The science of measuring colors is called 'colorimetry', and instruments are known as "colorimeters". When measuring color, the main task is to determine its coordinates, and this is carried out with the help of tricolor colorimeters or is calculated based on the spectrum of diffuse reflection or light transmission [27]. Colorimeters are devices that, by means of their light sources, emit a momentary beam of light, which is reflected on the surface of the tooth and is returned to the apparatus through the same opening from which they came out to be processed by the appliance [28]. The color elements of the obtained light rays are analyzed; thus, the color of the tooth surface is obtained [29]. Colorimeters comprise three main elements, a light source, a detector and filters. These instruments provide stable measurements on transparent and translucent objects, but are not reliable when measuring complex and multilayer translucent structures. Because they do not record all spectrally reflected visible light and due to the aging of the filters, they are less accurate than spectrophotometers [30].

- *Spectrophotometers*

Spectrophotometers can be defined as the most accurate, useful and applicable in everyday practice devices. They measure the amount of light energy reflected from an object at intervals of 1 to 2.5 mm for the entire visible spectrum [31]. These devices consist of a light source, a light scattering device, an optical measuring system, a detector and a device for converting the received light into a signal that can be subsequently analyzed. Compared to the visual method, spectrophotometers offer a 33% increase in accuracy, and in 93.3% of the cases, color matching. Their difference with colorimeters is that they analyze the scattering of the entire visible spectrum 400–700 ηm. This quality makes them slower than colorimeters [32].

- *Delta E (ΔE)*

The International Commission on Illumination determined the mean color change (ΔE). Delta E is a standard measurement created by the Commission Internationale de l'Eclairage (International Commission on Illumination). Delta E is defined as the difference between two colors in an L*a*b* color space. As the values determined are based on a mathematical formula, it is important that the type of color formula is taken into account when comparing the values. The CIE L*a*b* formula used in the proofing market calculates the Euclidian distance, i.e., purely the distance between two points in a three-dimensional color space. The actual position of the points themselves is irrelevant [33]. Delta E*ab or ΔE*ab was the first internationally endorsed color difference formula introduced by the International Commission on Illumination (CIE). It is a single number, calculated using the formula below, to determine the difference between two colors. The 1976 formula is the first formula that related a measured color difference to a known set of CIELAB coordinates. This formula has been succeeded by the 1994 and 2000 formulas because the CIELAB space turned out to be not as perceptually uniform as intended, especially in the saturated regions. This means that this formula rates these colors too highly as opposed to other colors (Equation (3)).

In case of the L*a*b* space, the $\Delta E_{Lab}$ difference between the two colors is calculated by the formula below, according to the formula for Euclidean distance between two points in the CIE L*a*b* space (Equation (3)).

$$\Delta E^*_{Lab} = \sqrt{\Delta L^{*2} + \Delta a^{*2} + \Delta b^{*2}} \tag{3}$$

Equation (3) is the first formula of $\Delta E_{Lab}$.

CIE94 introduced a conversion of the given Lab value into CIE L*C*h (Lch). The two colour models differ in that Lch represents hue as an angle instead of infinite points of color. This allows us to more easily troubleshoot and perform calculations on hue [34].

The aim of this paper is to review the available literature on the different methods for colour stability determination of the CAD/CAM milled and 3D printed resins for denture bases.

## 2. Materials and Methods

The methodology included applying a search strategy, defining inclusion and exclusion criteria, and retrieving studies; selecting studies; extracting relevant data to summarize the results. Searches of PubMed, Scopus, and Embase databases were performed to gather the literature published between 1 January 1998 and 31 March 2022. The search terms used were "CAD/CAM" [Mesh] OR "3D printing" OR "Milled acrylic resins" OR "3D additive manufacturing" AND "Subtractive manufacturing" [Mesh].

The inclusion criteria for selection were 186 articles written in English published between 1 January 1998 and 31 March 2022 on CAD/CAM milled acrylic resins for removable dentures, clinical studies and in vitro studies, articles that reported different methods for determination of the color stability, clinical performance or quality assessment with CAD/CAM milled dentures. This period of time, between 1 January 1998 and 31 March 2022, was selected particularly, due to the increase in clinical and in vitro studies during these years. Exclusion criteria included 76 articles that failed to involve items described in the inclusion criteria or described repetitive data already included.

## 3. Results

The search strategy for this review involved three stages, reviewing titles, abstracts, and final selection of articles for full text analysis. A total of 82 from the other 110 articles selected from the databases were sorted independently by 3 reviewers, and any differences in selection were discussed until a consensus was reached. Upon the reviewers' agreement, articles that did not meet the predetermined inclusion criteria were excluded. Abstracts of the articles selected at the second stage were independently evaluated by the same reviewers, and 66 of these 82 articles were selected for the final analysis and were obtained in full text. At the third and final stage, the full text of the obtained 66 articles was analyzed.

In the study of Oguz et al. [35], a spectrophotometric evaluation was carried out to measure color difference, to reduce the chances of subjective error through the visual method. Based on the data obtained through the spectrophotometric evaluation, the hypothesis tested in the study was rejected. After 30 days immersion in staining solution, coffee caused a slightly higher change in color then tea and coke. However, the results of the present study oppose the findings by Buyukyilmaz et al. [36], who found that all the materials used in their study were at same discoloration level after 96 h of immersion in coffee and tea solution. Similarly, in a study carried out by Um and Ruyter et al. [37], they demonstrated that coffee solution causes less discoloration of material then tea after 48 h of storage of five resin-based veneering materials. According to Lai et al. [38], the hydrophobic silicon material in the resin-based material was more deeply stained by coffee solution than the tea solution. Hydrophobic staining solutions stained with hydrophobic materials more easily.

Perceptibility and acceptability are the most often used threshold units in the field of color science. Such thresholds can be very significant to color quality control processes in

the printing industry and should be defined in quality measure units. Optical density (OD) and color difference models are usually utilized as color quality measures and have been provided in a considerable number of commercial measuring devices, such as spectrophotometers. Kim et al. [39] measured the acceptability sensitivity of human observers to the reflection OD variation using a simpler method of limits and proposed a computational model for predicting the acceptability threshold and sensitivity as a function of OD of given color patches. Their findings show that the OD threshold function can be explained by a log function of OD. The acceptability and the perceptibility thresholds across the reflection OD domain were psychophysically measured using the method of limits. The former steadily increases as the OD increases from the low to high density fields and it rapidly decreases at higher density fields [40].

## 4. Discussion

Various factors may affect the color change in denture base materials after prolonged use. These factors are as follows: water absorption, stain accumulation, degradation of intrinsic pigments, dissolution of ingredients, foods, beverages and roughness of surface. Color change in dental materials is clinically very important for the dental operator, as it determines the clinical serviceability of the material. Changing the color of prosthetic materials can lead to discomfort and dissatisfaction in the patient and additional replacement costs [41].

In a comparative study, both prosthetic bases and artificial acrylic teeth were stained. To determine the effect of daily consumed beverages (e.g., tea, coffee and Pepsi®), a color stability study of two types of heat-curing acrylic plastic prosthetic bases (Hiflex-H, Prevest DenPro®, USA and DPI® Heat Cure, Dental Products of India, India) was performed [42]. The largest change in color was evident among the tea samples after 30 days ($\Delta E = 39.21$), compared to samples immersed in distilled water ($\Delta E = 1.43$). Tea immersion led to the largest changes in color according to the National Bureau of Standards (NBS), followed by Pepsi® and coffee. Increasing the immersion period also increased the staining [43,44]. Studies on color changes in microwave polymerizing resins by immersion in different beverages observed that there is no difference in color when different polymerization regimes were applied [45–47].

While denture base resins may undergo color changes over time due to intrinsic and/or extrinsic factors, extrinsic factors are considered the more frequent causes of color changes. The aim of this study was to assess the color stability of denture base material using three extrinsic staining beverages (tea, coffee, and Pepsi) and one control solution (distilled water) [48]. The highest color change was apparent with tea after 30 days when compared to specimens stored in distilled water. This result is in agreement with the study of Hatim et al. [49], where they found tea caused the highest color change compared with coffee and Pepsi.

Other studies conducted by Turker et al. [50] and Waldemarin et al. [51] found the same result. The second-most discoloring beverage in the present study was Pepsi®, followed by coffee. The present authors' result, however, is not comparable with some other studies in which the highest color change was observed with coffee over tea and Pepsi®. One study conducted by Amin et al. found that Coca-Cola® was the most discoloring beverage over coffee and tea [52].

Increasing the immersion time of specimens in the discoloring beverages caused an increase in the staining of the acrylic resin denture base material, as already reported in the literature [53,54]. Dentists should take the initiative to increase patient awareness about discoloration with certain beverages that might affect the denture base and potentially lead to additional expense for replacement. [55–57] Patients may, for example, wash their mouths and clean their dentures after drinking staining beverages using tap water or any commercial mouthwash to minimize the discoloration effect [58,59]. Dental laboratories/suppliers should use high-color-stable acrylic resin denture base materials to ensure high-quality dental service to patients [60,61].

According to Dayan et al. [62], the color stability of CAD-CAM denture base resins is better than any of the other kind of denture base resins. In this study, 60 disc-shaped specimens were randomized into the following 4 groups (n = 15) according to storage media: coffee, coke, red wine and distilled water (control group). The color measurement of each sample was performed using a spectrophotometer before and after storage (after 7 and 30 days), and color changes (ΔE) were calculated. All the denture base materials demonstrated dissimilar color changes after stored in the different storage media in both evaluation stages. In all storage media, CAD-CAM denture base resins showed the minimum color change. In all denture base resins, red wine showed a higher degree of color change than coke or coffee.

Another study conducted by Alp et al. [63] examined the effect of coffee solutions on the discoloration of different CAD-CAM acrylic resins; similar to the current study, researchers reported that clinically admissible color changes did not occur in different denture base acrylic resins due to coffee staining. However, researchers observed that the color change and surface roughness in heat-polymerized and different pre-polymerized CAD-CAM PMMA specimens were not significantly different. This may be attributed to the 8 h of heat polymerization of the heat-polymerized control group, which enhance its physical features.

In the study of Papathanasiou et al. [64], the two-way ANOVA revealed that the interaction between material and staining solution significantly affected color changes after immersion ($F_{(9,96)}$ = 44.67, $p < 0.001$). The polyamide materials had the highest color change overall (ΔE* = 14.59 ± 8.65) ($p < 0.001$). Coffee caused the highest color (ΔE* = 13.08 ± 6.98) and gloss changes (ΔG = −6.36 ± 19.2 GU) among different solutions ($p < 0.001$). PEEK showed the highest alteration of gloss (ΔG = −11.31 ± 15.49 GU), with significant difference with the other three materials ($p < 0.001$). Insignificant interaction of the material and immersing solution was found for surface roughness parameters ($p > 0.05$).

Zuo et al. [65] examined the discoloration of different denture base resins after immersion in different cleaners and different beverages. The conclusion of the study points to the fact that color change in the Eclipse denture base resin was much higher than the clinically acceptable value of ΔE 3.3. This is in line with the present study results, which showed that the Eclipse denture base groups had the most defined color change. This result could be due to the tendency for high water absorption in light activated denture base materials when compared to the other materials. Kerby et al. reported that Eclipse is also sensitive to hygroscopic expansion; this is caused by the two hydrophilic urethane groups within its molecular structure, but less than PMMA [66].

Similarly, Shin et al. [67] evaluated the color stability of CAD/CAM blocks and 3D-printing resins for their degree of discoloration based on material type, colorant types, and immersion duration in the colorants. The authors concluded that 3D-printing resins demonstrated color differences above the AT (ΔE > 2.25) following immersion for 7 days or longer in all test groups. The authors also revealed that after thermal cycling, the water sorption of 3D-printed resin was high compared to prefabricated PMMA. However, in the same study, the authors showed that each study material showed distinct properties, even when using the same 3D-printing method. Thus, it was evident that other factors, such as material properties and other output parameters, could affect the water sorption rate of 3D-printing resin. While water sorption alone does not justify low color stability, it could be considered a contributing factor [68].

## 5. Conclusions

The infusion of CAD/CAM techniques into CD (complete dentures) fabrication methods has led to the evolution of modified and easier clinical protocols, the use of materials with improved properties, satisfactory color stability, better fit and retention of the CDs and reduction in the chairside and laboratory times. Discoloration of the denture bases can be evaluated with various instruments. Since instrument measurements eliminate the subjective interpretation of visual color comparison, spectrophotometers and colorimeters

have been used to measure color change in dental materials. Various studies have reported different thresholds of color difference values above the color change perceptible by the human eye. Additively manufactured denture resins demonstrated the maximum color change compared to conventional heat-polymerized and CAD/CAM subtractively manufactured denture resins. All beverages used in the studies had an effect on color change. The following conclusion could be drawn: the color stability of CAD-CAM milled and 3D printed resins is not inferior to the conventional type of acrylic resins for denture bases.

**Author Contributions:** Conceptualization, M.D. and M.C.; writing—original draft preparation, M.D. and M.C.; writing—review and editing R.K., A.V., G.B., F.D. and Z.T.; supervision, S.K. and S.C. All authors have read and agreed to the published version of the manuscript.

**Funding:** This research received no external funding.

**Institutional Review Board Statement:** Not applicable.

**Informed Consent Statement:** Not applicable.

**Data Availability Statement:** Not applicable.

**Conflicts of Interest:** The authors declare no conflict of interest.

## Abbreviations

| | |
|---|---|
| PMMA | Polymethyl methacrylate |
| STL | Stereolithography, standard triangle language, standard tessellation language |
| 3D | Three-dimensional |
| CAD/CAM | Computer-aided design/computer-aided manufacturing |
| CD | Complete dentures |
| CIE | International Commission of Illumination |
| NBS | National Bureau of Standards |
| OD | Optical density |

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
