# Peer review of "Color Stability Determination of CAD/CAM Milled and 3D Printed Acrylic Resins for Denture Bases: A Narrative Review"

_jcs, doi:10.3390/jcs6070201_

Round 1

Reviewer 1 Report

You can find attached the PDF file.

Only average 15 articles are described in this article above the 64 found in the electronics databases.

Author Response

Dear Reviewer,

We really appreciate your time and effort. We will comply with your suggestions and will make the necessary corrections. Thank you very much!

Reviewer 2 Report

The authors present a narrative review aiming to describe the methods available for color determination of CAD/CAM acrylic resins used for denture bases.

The authors begin with an introduction describing CAD/CAM, followed by a description of how the review was performed. After they describe the color properties and how they can be evaluated. The last section details studies evaluating color changes in CAD/CAM acrylic resins.

Lines 26-32: these sentences are not related to the manuscript's aim and title since they describe the CAD/CAM methodology and its advantages without any reference to the color stability of the materials.

Line 46: please refer to the meaning of PMMA before using the abbreviation. Also, although being the most documented, PMMA is not the only available resin for denture bases. So why do you only refer to PMMA?

Materials and methods: the authors present a materials and methods section as this was a systematic review instead of a narrative one. Why did you decide to do that? If you performed the review as a systematic review, why did you not present it that way?

Section 2: the number of included and excluded articles should be moved to the results section.

Line 68: the publication language is not an inclusion criterion but a filter you apply during your literature search. Please correct it.

The authors performed a search from 1998-2022 but only included articles between 2019 and 2022. Why?

Lines 70-71: If you intended to evaluate the color stability, why did you define as inclusion criteria studies that evaluated clinical performance or quality assessment?

Line 73: what do you mean by “described repetitive data already described”? How was this evaluated?

Sections 3.1, 3.2 (3.2.1 and 3.2.2), and 3.3 are not review results but information that should be placed in the introduction section since it is not relevant results related to the review aim.

Section 3.2.1: I suggest including the description of translucency here (lines 159-167), so the four dimensions are described together.

The visual methods described for color determination apply to all dental materials, not just CAD/CAM acrylic denture bases. So, again, this information is not a review result, but again, information that should be placed in the introduction section.

The review results are described in sections 3.3.3 and the discussion section. Therefore, the information presented in the discussion section should be moved to the results. Importantly, the information presented should be complemented since important information is missing, for instance, the materials tested (type, composition…)

The manuscript lacks an appropriate discussion on the methods available for studying the color stability, their limitations, and which factors influence color stability…

Lines 266-269: what study is being referred to?

Conclusion: please shorten this section.

Plagiarism software detected several problems with a previously published paper from the same authors. Please correct them.

Unfortunately, overall, I find the manuscript confusing and hard to follow. It lacks a clear focus on what is being evaluated and described since most of the manuscript focuses on describing information about CAD/CAM and color properties, which is not the review aim. The review results are mostly presented in the discussion section and are incomplete. I recommend the authors rearrange and complete the given information and write a complete results section, describing the different studies, how the color stability was evaluated in each one and their main conclusions.

Author Response

Dear reviewer,

We really appreciate your time and effort. We will comply with your suggestions and will make the necessary corrections. Thank you very much!

Reviewer 3 Report

Dear authors

Thank you for an interesting report.

In this study, you reviewed color stability determination of CAD/CAM milled acrylic resins for denture bases. In recent years, the rapid development of digital dentistry based on CAD / CAM technology can be expected to solve various problems that have been encountered in dentistry so far. Regarding the production of denture bases, there are many advantages from the viewpoint of reproducibility and work efficiency, so it is hoped that related research will be deeply. Therefore, it is thought that this review report will be useful for many readers of dental field such as dental technicians, prosthodontic researchers, and clinical dentists.

I agree to many parts of your claims and guessed that the subject of this paper will be of interest to the readership of this journal. However, I think that minor revisions are required as follows:

Pages 1-2, 1. Introduction

1. Although the advantages of CAD / CAM and CAD / CAM milled acryl resin and the disadvantages of acrylic resin are described, the disadvantages of milling acrylic resin blocks with CAD / CAM technology to make denture bases, such as existence of cutting chips after milling. Is not written at all. I think some kind of description is necessary.

Page 2, Materials

1. The 65th line

I think that you should describe why the papers from 1998 to 2022 were targeted when searching for published literature. Perhaps it is because there were no published papers before that, but you should still state it clearly.

2. The 69th line

Similar to above (1), I think that you should describe why the papers from 2019 to 2022 were targeted when searching for published literature.

Throughout the manuscript

1. CAD / CAM and CAD-CAM are mixed throughout the article. If for some reason the wording is different, the change is not needed, but if not, I think that you had better unify it to the CAD / CAM wording. If this difference in expression is normal, please ignore my point.

Author Response

Dear reviewer,

We appreciate your time and effort. We will comply with your suggestions and will make the necessary corrections. Thank you very much!

Round 2

Reviewer 1 Report

You can find attached the PDF file.

Author Response

We will comply with your suggestions. Thank you very much for your help!

Reviewer 2 Report

The authors present a narrative review aiming to describe the methods available for color determination of CAD/CAM acrylic resins used for denture bases.

The performed modifications improved the manuscript quality and clarity, but further corrections are needed.

Lines 29-32: I recommend you only refer to color stability and not to the other properties that are unrelated to the study aim.

Line 45: please refer to the meaning of PMMA before using the abbreviation.

Lines 102-110 and 116-125: I suggest these sentences be moved and placed after line 83 since section 1.1.1 is where the color dimensions are first described.

Since this is a narrative review, I suggest removing the materials and methods section and the first part of the results (lines 233-256). I think the manuscript becomes clearer and easier to follow without these sections.

Lines 243-244: if the authors decided to maintain this section, I suggest these criteria be removed since they are unrelated to the study aim.

Again, I think the results of studies evaluating color changes in acrylic resins should be moved from the discussion to the results section.

The manuscript lacks an appropriate discussion.

Author Response

(The authors gave the same response as above.)

Reviewer 3 Report

June, 22, 2022

Dear authors

Thank you for an interesting report.

After reviewing your report again, I found it much easier to understand because the points I pointed out were improved.

I agree to your claims and guessed that the subject of this paper will be of interest to the readership of this journal.

Author Response

Thank you very much for your help and support!

Round 3

Reviewer 1 Report

Lines 40-42: "imported in .STL (stereolithography, standard triangle language, standard tessellation language) format, which segments CAD drawings into small thin sections, thus allowing processing them layer by layer. [2]" This sentence refers too much to additive manufacturing. Title also include milling!!

Line 54: "and porosity[5] Another" A point must be added.

Line 264: Reference number must be checked.

Reviewer 2 Report

The authors present a narrative review aiming to describe the methods available for color determination of CAD/CAM acrylic resins used for denture bases.

The performed modifications improved the manuscript quality and clarity, but the manuscript's main problems remain: the methodology and results section.

Manuscript: the meaning of CAD/CAM and PMMA abbreviations need to be referred to. The abstract is an independent section, and the abbreviations' meaning needs to be referred to both in the abstract and the manuscript body.

Lines 79-90: I recommend this information be placed after lines 105. The color dimensions are only described in line 91, so it makes no sense to describe the dimensions before they are referred to in the first place.

Figures 1 and 2: are these figures original or reproduced under permission? If so, a reference from the source of the figures must be added to the figures' caption.

Line 281: the number of excluded articles should be moved to the results section.

The authors refer to 66 included studies, but the results section refers to 6 studies' results. What about the 60 others? Are they not relevant? Even considering the studies in the discussion section, some studies are still not referred to.

Lines 277-279: if the authors decided to maintain this section, I suggest these criteria be removed since they are unrelated to the study aim.

The manuscript lacks an appropriate discussion.

This manuscript is a resubmission of an earlier submission. The following is a list of the peer review reports and author responses from that submission.

Round 1

Reviewer 1 Report

  1. In the title appears  "milled acrylic resins", it was not considered as it is in the search criteria. CAM acrylic resins can be also printable, s.a.
  2. how many articles finally met the search criteria used? where are the results clearly described?
  3. composition and manufacturing of CAD/CAM blocks is important, and can be relevant to colour stability perspective, but should be discussed after results found. 

Reviewer 2 Report

The main problem of this review article is the fact that topic is extremely narrow and I have impression that it was created "artificially" only to prepare this work.

In the introduction or in the text itself, it is not explained why testing methods for measuring the color of these materials should be reviewed? Is there anything unique here that would suggest a specific methodological approach for CAD / CAD materials compared to materials processed by other methods?

As a result, in the content of the work we have a whole range of basic information available in many works (where it is much better developed), which could as well be in the introduction (e.g. 3.1). Moreover, most of the cited works do not even apply to CAD / CAM materials (moreover, the overwhelming majority of them are not, as the authors declare, works published between 2019 and 2022) ....

Chapter 3.2 describes the basic methods of color measurement. There is no account here that would indicate that the considerations apply to CAD / CAM materials. The description is absolutely basic to the message.

In the following chapters there is a similar approach and chaos: for example, in Section 3.3 the authors write about the choice of teeth color, when the title of the work suggests that it concerns denture base materials. Section 3.3.3 provides general information on the discoloration of a wide variety of materials. The discussion section is very general and is based on just 14 papers. Most of this work concerns other materials, and some are very old. CAD / CAM materials were mentioned in three cited works, but in two were investigated - the third is a review article of the authors of the currently evaluated work.

In my opinion, the work is chaotic, very general and does not really concern the discussed topic. I am very sorry but I do not see any reasons to publish it.